# An Innovative Preparation, Characterization, and Optimization of Nanocellulose Fibers (NCF) Using Ultrasonic Waves

**DOI:** 10.3390/polym14101930

**Published:** 2022-05-10

**Authors:** Abdullah K. Alanazi

**Affiliations:** Department of Chemistry, College of Science, Taif University, P.O. Box 11099, Taif 21944, Saudi Arabia; aalanaz4@tu.edu.sa

**Keywords:** grass nanofiber cellulose (GNFC), NaClO, NaOH and H_2_SO_4_, crystallinity index, Zeta potential, XRD, SEM, particle size distribution

## Abstract

Recently, environmental and ecological concerns have become a major issue owing to the shortage of resources, high cost, and so forth. In my research, I present an innovative, environmentally friendly, and economical way to prepare nanocellulose from grass wastes with a sodium hypochlorite (NaClO) solution of different concentrations (1–6% mol) at different times 10–80 min, washed with distilled water, and treated with ultrasonic waves. The optimum yield of the isolated cellulose was 95%, 90%, and 87% NaClO at 25 °C for 20 min and with NaOH and H_2_SO_4_ at 25 °C with 5% M, respectively. The obtained samples were characterized by dynamic light scattering (DLS), Fourier-transform infrared (FT-IR) spectroscopy, and X-ray diffraction (XRD). The effect of test temperature and reaction times on the crystallinity index (I_C_) of GNFC with different treated mediums was carried out and investigated. The I_C_ was analyzed using the diffraction pattern and computed according to the Segal empirical method (method A), and the sum of the area under the crystalline adjusted peaks (method B) and their values proved that the effect of temperature is prominent. In both methods, GNFC/H_2_SO_4_ had the highest value followed by GNFC/NaOH, GNFC/NaClO and real sample nano fiber cellulose (RSNFC). The infrared spectral features showed no distinct changes of the four cellulose specimens at different conditions. The particle size distribution data proved that low acid concentration hydrolysis was not sufficient to obtain nano-sized cellulose particles. The Zeta potential was higher in accordance with (GNFC/H_2_SO_4_ > GNFC/NaOH > GNFC/NaClO), indicating the acid higher effect.

## 1. Introduction

Nanocellulose presents a significant achievement in science and technology, due to its regular atomic arrangement, stiffness comparable to steel, and successful use as an organic filler material in biopolymer nanocomposites, [1]. Moreover, Blessy et al. [2] stated that cellulose has low cytotoxicity, biocompatibility, good mechanical properties, high chemical stability, and cost effectiveness, which makes it a suitable candidate for biomedical applications. Zou et al. [3] designed and fabricated a mussel-inspired, low-cost, polydopamine-filled cellulose aerogel with both super hydrophilicity and under water super oleophobicity. Zhenghao et al. [4] concluded that cellulose, lignin, and lignocellulose not only protect the environment but also reduce dependence on fossil resources. Kusmono et al. [5] used ramie fibers accompanied by sulfuric acid hydrolysis, with a high crystallinity (90.77%), small diameter (6.67 nm), and length (145.61 nm). Lu et al. [6] applied the ultrasonic wave and microwave-assisted technique (SUMAT) for the preparation of nanocellulose. Syafri et al. [7] used the solution casting method for fabricating nanocellulose from water hyacinth (*Eichhornia crassipes*).

Chen et al. [8] showed that XRD profiles of Cr(NO) hydrolysis to isolate cellulose nanocrystals CNCCr(NO) had major peaks at around 2θ = 22.50° (200). Sangeetha et al. [9] showed intensive peaks at 2θ = 16.39°, 20.62°, and 22.60°, and Garvey et al. [10] showed XRD peaks at 15.00°, 22.50°, and 34.85°. Ju et al. [11] mentioned that the crystalline and amorphous peaks deconvolution on which the crystalline cellulose is represented by several intense peaks at (1ī0), (110), (102), (200), and (004). Yazdani et al. [12] showed that the amorphous subtraction method used to fit the amorphous component intensity profile. Agarwal et al. [13] and Segal et al. [14] calculated I_C_ by subtracting the amorphous contribution approximately at 2ϴ = 18° as follows:I_C_ = 100 × [(I_22_._5_° − I_18_°)/I_22_._5_°].

Khukutapan et al. [15] used autoclaving cabbage outer leaves for the production of nano-fibrillated cellulose with a crystallinity index (CI) of 50.70% and cellulose content of 49.20% dry mass. Hu et al. [16] isolated fibrils from bamboo fiber (BF) with the assistance of negatively charged parts with the yield above 70.00% using the ultrasonic homogenization. Barbash et al. [17] treated bleached softwood sulfate pulp mechanochemically. Barbash et al. [18] prepared nanocellulose from organosolv straw pulp (OSP). Thakur et al. [19] concluded that cellulosic waste features dependents on the method of extraction. Sharma et al. [20] found that the cellulose is the biosynthetic product of plants, animals, and bacteria. Chen et al. [8] reported feasibility and practicability of the hydrolysis using CNCCr(NO) from native cellulosic feedstock that exhibited a higher crystallinity (86.50% ± 00.30%) and high yield (83.60% ± 00.60%). Mazela et al. [21] attempted to evaluate hybrid cellulose treatment, using a combination of a chemical method and ultrasound of medium frequency. Zhang et al. [22] showed that when the bagasse nanocellulose was rod-like and its content in PHB was 1 wt.%, the toughness of PHB (polyhydroxybutyrate) composite was the best. Park et al. [23] illustrated the effect of cellulose crystallinity on its accessibility, lignin/hemicellulose contents and distribution, porosity, and particle size. Chargot et al. [24] obtained nanocellulose from apple pomace. Trache et al. [25] illustrated the recent advances in the nanocellulose preparetion. Thomas et al. [26] used *Acacia caesia* fiber for the isolation of nanocellulose whiskers. Barbash et al. [27] used *Miscanthus giganteus* stalks to make organosolvent pulp and nanocellulose that has a crystallinity index of 76.50%. Yahya et al. [28] found that oil palm (*Elaeisguineensis*) empty fruit bunch (OPEFB) has nanocellulose yield of 81.37%. Duan and Yu [29] concluded that the jute fibers nanocellulose has high yield of 80%. Ma et al. [30] extracted nanocellulose from *Xanthoceras sorbifolia* husks through a series of chemical treatments, after which the obtained nanocellulose had a rod-like shape and diameter of 38 nm.

The present work aimed at investigating the effect of sonicated NaClO medium on the production of GNFC at different concentrations, test temperatures, and reaction time. The effect of sonication on GNFC/NaClO, GNFC/NaOH, and GNFC/H_2_SO_4_ systems in aqueous media on the crystallinity index (I_C_) was evaluated at different medium concentrations, test periods, and reaction times. SEM, XRD, FT-IR, HPLC, and Zeta potential technique were used. The size distribution of GNFC/NaClO, GNFC/NaOH, and GNFC/H_2_SO_4_ as measured by particle size analysis was obtained.

## 2. Experimental

### 2.1. Materials and Chemicals

Garden grass cellulose fibers were isolated and mowed in large quantities periodically from the vegetation cover of large areas of stadium floors, parks, and public gardens followed by 90.00% ethanol treatment at 70 °C. To isolate fibers, pigments, dusts, and fats were removed from the purified grass by washing with water. Bleaching with 120 mL of household bleaching agent (5% NaClO and 5% NaOH), carried out followed by drying. The product mixed with acetic anhydride (100 mL), glacial acetic acid (100 mL), and sulfuric acid (10 mL) and cooled to 7 °C 35 g used.

### 2.2. Preparation of KGNFC (Alkali Grass Nanofiber Cellulose)

The grass rinsed with NaOH solution, water, and finally with H_2_O_2_ (hydrogen peroxide) 3 times consecutively. As H_2_O_2_ dissolved hemicellulose, the color turned from green to white leaving only the cellulose in the mixture and it had 97.00% purity after drying. Sulphuric acid (98.00% concentration) was added to cellulose colloid (5 g cellulose powder + 250 mL water) under constant stirring condition for 3 h. 

The suspension produced heated at 50 °C for 2 h and diluted 10 times with distilled water ice-cooled to prevent the acid hydrolysis reaction. The resulting white colloidal suspension centrifuged at 8000 rpm for 20 min followed by dialysis to remove excess acid and ultra-sonication for 45 min in ice cooled condition and lyophilized [31].

### 2.3. Preparation CGNFC (Acidic Grass Nanofiber Cellulose)

The fibers purified fibers hydrolyzed using 5% sulfuric acid at different conditions using two parameters (temperature and hydrolysis time) with H_2_SO_4_ of a constant concentration of 5 wt.% under different temperatures of 25 °C, 40 °C, 60 °C, and 80 °C and 10, 20, 40, 60, and 80 min to get the optimum temperature and time with high yield %. Centrifugation was conducted at 4500 rpm for 20 min to abolish the acid solution. Cellulose nanofiber (CNF) precipitates were collected and rinsed with distilled water to neutral condition. Then the ultra-sonication of CNFs suspension performed for 20 min and 50% amplitude to obtain the uniform cellulose nanofiber (CNF) suspensions [14,32,33].

### 2.4. Innovative Preparation of Grass Nanofiber Cellulose (IGNFC)

The collected grass was washed with rinsed water and ground mechanically with sodium hypochlorite of different concentrations, 1, 1.5, 2.5, and 5.0 mol, for 10, 20, 40, 60, and 80 min at 25 °C, 40 °C, 60 °C, and 80 °C. After that, the product was repeatedly washed with rinsed water and treated with ultrasonic waves, and the size of the particles was determined, (Figure 1). All methods of GNFC preparation methods are compared by studying particle size and yield using LDS, SEM, XRY, HPLC, and EDX methods and Zeta potential analysis.

#### 2.4.1. Characterizations

##### Fourier Transform Infrared (FT-IR) Spectroscopy

Perkin-Elmer RX X2 Infrared spectrometer spectra used for RSNFC, GNFC/NaOH, GNFC/NaClO, and GNFC/H_2_SO_4_ using 4500‒500 cm^−1^ wave range, 1 cm^−1^ intervals and 4 cm^−1^ scanning resolution. 

##### X-ray Diffraction (XRD)

Four samples were prepared from untreated grass fiber, and three synthesized nanocellulose: NAFC (nano acid cellulose), NKFC (nano alkaline cellulose), and NSFC (nano sodium hypochlorite cellulose). Three pellets were prepared, and measured in reflection mode, in the range 2θ = 5°–80° with Philips powder diffractometer with Cu Kα radiation (k = 0.154 nm), using Ni-filtered Cu Kα radiation (λ = 1.5406 Å) at 40 kV and 30 mA [34]. XRD studies were performed to evaluate the effect of each treating medium on the crystallinity behaviors of native cellulose and yielded nanocellulose specimens. The (I_C_) calculated using two methods, the first (A method) according to the Segal empirical method [14] as in Equation (1):CrI% = [(I_200_ − I_am_)/I_200_] × 100 (1)
where, I_200_ is the crystallites peak intensity at 2θ = 22.5° and I_am_ is the amorphous cellulose intensity at 2θ = 18°–19°.

The second approach (B method) is a deconvolution according to [8,35] as in Equation (2):I_C_% = [A_cryst/(_A_cryst_ + A_amorph)_] × 100(2)
where A_cryst_ is the calculated area under X-ray and A_amorph_ is the total area under the X-ray pattern. The crystallite size (t) calculated according to Scherrer equation, [14]:t = [Kλ/(β_1/2_ cos ϴ)](3)
where K = 0.89 is Scherrer constant, λ = 1.54060 Å is the radiation wavelength, β1/2 equal the full width at the half maximum (FWHM) of (200) diffraction peak in radians, and θ is the corresponding Bragg’s angle.

##### SEM Analysis

The cellulose microstructure morphology examined using a JEOL JSM-7001F TTLS (JEOL Ltd., Tokio, Japan) SEM with 5 kV accelerating voltage of 5 kV, 10 mm working distance of about 10 mm.

##### Determination of GNFC Yield

The oscillating ultrasonic frequency used was 40 kHz with an output of ultrasonic power of 40 KW. The nanocellulose yield was calculated according to the relation (4) [16,17,18,19];
Y = {[(m_1_ − m_2_) × V_1_]/mV_2_} × 100%(4)
where Y is the yield of GNFCs, m_1_ and V_1_ are the GNFC mass and volume and weight bottle, m_2_ and V_2_ are the mass and volume bottle weight. m is the mass of grass fibers. 

##### High-Performance Liquid Chromatography (HPLC)

HPLC was carried out using (HPLC, Shimadzu Corp., Kyoto, Japan). With a rotary shaker agitation of 25 mL of GNFC solution added in a flask with 50 mg GNFCs, the suspension produced filtered using a membrane filter (Millipore 0.45 lm pore size). The filtrates were analyzed for residual GNFC. HCl used to adjust the solution’s pH. The absorbed NC (nanocellulose) centrifuged with an initial dose of 2.0 g/L, water washed, and dried for 24 h at 35 °C. The NCs product conducted with various eluents such as 5% H_2_SO_4_, 5%NaOHand 5% NaClO by repeating the above procedure for two times to be used with HPLC [36,37].

##### Zeta Potential Measurement

Zeta potential used a Zetasizer Nano series for determining the electrophoretic mobility according to Henry equation [38]. 

## 3. Results and Discussion

### 3.1. Temperature Effect on the GNFC Yield

The effect of reaction temperature of 20 °C, 40 °C, 60 °C, and 80 °C on GNFC yield shown in Figure 2. The optimum NaClO/GNFC yield value was 95% at 5% M, 20 min, and 25 °C. Figure 2a–d, indicated that the GNFC yield increased first followed by continuous decrement until reaching a minimum value about 38% at 80 °C, 80 min, with 6 M of NaClO due to the effect of both the higher temperature and hydrolysis medium on removing the amorphous components and accelerating the glycosidic bonds.

Furthermore, this is related to the fact that both the higher temperature, and treated hydrolysis medium hydrolysis removed amorphous components and some parts of crystalline that accelerating the hydrolytic cleavage of the glycosidic bonds and finally resulted in the yield and crystallinity decrement [5,16,38,39].

### 3.2. Time Effecton the GNFC Yield

Figure 2a–d showed that the GNFC yield decreased from 95% at 20 min, 25 °C of 5 M NaClO, to 38% at 80 min, 80 °C, and 6 M NaClO. Up to 20 min, the GNFC yield value increased directly due to the specific surface increment and more cellulose depolymerization followed by yield value decrement owing to the ultrasonic wave effect. Above 20 min, the yield decreased owing to the crystalline cellulose hydrolysis, [16]. So, 20 min us considered as optimum time.

### 3.3. Effect of NaClO Concentration

Figure 2a–d revealed that when NaClO concentration increased from 1.0 to 6.0 M, the nanocellulose progressively increased from 87% to 95% at 25 °C and 20 min due to the Effect of catalytic hydrolysis process that causes crystallinity increment. Cellulose levels decreased with the increase of NaClO concentration over 5.0 M at longer time exposure at high temperatures since NaClO solution increment will fractured both the hemicellulose and cellulose connection ties. In an alkaline solution (NaClO), the temperature increment causes the lignocellulosic components destruction and the bonds termination in agreement with Winarsih [40].

### 3.4. Effect of Reaction Medium on GNFC Yield

Figure 3 illustrates the dependence of reaction medium on both reaction temperatures and reaction time. Three reaction mediums were used: NaClO, NaOH, and H_2_SO_4_. Figure 3a indicates that the optimum yield temperature was 25 °C with 95%, 90%, and 87% for 5% concentration of NaClO, NaOH, and H_2_SO_4_ respectively. The yield values of the three mediums increased up to 25 °C followed by a yield decrement with temperature increment, since high temperatures reduced the reaction activity of cellulose with excessively hydrolyzed into glucose monomers, which reduced the yield of GNFC, in agreement with [16]. Also, with temperature increment; the power applied reduced the reaction activity of cellulose, so the mass transfer of the intra-finer pores of GNC is also reduced that result in the decrement of GNFC Yield with temperature increment in agreement with Lu et al. [6].

Figure 3b shows that the yield rises up to an optimum value of 20 min reaction Time for 5% concentration of each of the three mediums. When the reaction time increases more than 20 min, the mass transfer resistance gradually decreases and the specific surface of the grass wastes increases with reaction time increament owing to the breakdown of their net structure and even the enlargement of the interand intra-fiber pores, which results from the hydrolysis of more and more cellulose that leading to the decrement of GNFC Yield with reaction time increment, [6]. The yield decreased with time increment than 20 min, therefore, the optimum yield value for obtaining cellulose from NaClO/GNFC was 95% which is higher compared to the cellulose yield of NaOH/GNFC of the previous publications which was 89% of pineapple [41], 83.40% of blenched fiber [42], 67.40% for non-woody biomass constitutes [34], 85.40% for non-woody biomass constitutes, 54.30% for organo-solvent miscanthus pulp (OMP) [31], and 81.00% and 54.00% from flax fibers and cotton linters [29]. Moreover, the yield of NaClO/H_2_SO_4_ cellulose was 90.00% the yield of the previous investigators which was 82% for non-woody biomass constitutes [43], ranging between 55 and 60% for bleached kraft pulp of loblolly pinewood [44], 85.75% for filter paper [6], 83.60% for native cellulosic feedstock [45] (Appendix A), 84.00% for oil palm (*Elaeisguineensis*) empty fruit bunch [46], from flax fibers (81.00%) and 54% from flax fibers (81.00%) and cotton linters [14]. In addition, Figure 3a,b show that the temperature increment had a higher effect on the yield decrement compared to the reaction time [9]. 

### 3.5. X-ray (XRD) Diffraction Patterns Analysis

The XRD diffractogram profiles for all the test specimens are shown in Figure 4. It was clearly observed that the XRD patterns of the four specimens were similar and these slight shifts in angles and peaks proved the occurrence of cellulose in agreement with Chen et al. [8] and Maia et al. [47].

### 3.6. Effect of Additives on the Crystallinity Index (I_C_) of GNFC

The crystallinity index was analyzed using the diffraction pattern and computed according to the Segal empirical method (method A) [48] and the sum of the area under the crystalline adjusted peaks method (method B) [35]; their results are presented in Table 1 and Table 2. The effect of temperature on the crystallinity index of Real Sample (RSNFC) was found to range from 23.205% to 40.705% and 19.724% to 34.559% for method A and B, respectively, while the effect of reaction time on the crystallinity index of Real Sample (RSNFC) was found to range from 21.401% to 39.701 and 17.763 to 32.952 for methods A and B, respectively. This proves that the temperature effect is higher compared to the reaction time and that the I_C_ values based on method A are over that obtained by method B, in agreement with the previous investigators [11,12,38]. Furthermore, Table 1 and Table 2 indicate the crystallinity increment due to the dissolving of hemicellulose and lignin that causes the chemical purification increment [1,49,50]. Khukutapan et al. [15] and Cherian et al. [50] concluded that the alkali hydrolysis and bleaching causes the separation of the structural linkages between lignin and carbohydrates that leads to significant lignin and the GNFC crystallinity index increment to around 70.29%. However, Hu et al. [16] showed that the crystallinity was not significantly affected by post chemical modification of bamboo fiber at a low degree of substitution carboxymethylation (CM) stage. Barbash et al. [18] concluded that the hydrolysed and sonicated methods of cellulose increased the package ordering of the macromolecules due to the decreament of amorphous cellulose parts ratio which results in the increment of the I_C_ of the initial cellulose from 75.00%, 78.30%, and 79.80%, respectively. Both Barbash et al. [19] and Sánchez et al. [51] used the organosolv straw pulp (OSP) but Barbash et al. [19] had higher I_C_ which was 72.50%. Chen et al. [8] and Chen et al. [52] found that I_C_ for native cellulose, GNFC/H_2_SO_4_, and CNC Cr(NO) were 65.70%, 81.40%, and 86.50%, respectively. Zhang et al. [22] found that acid treatment causes the hydrogen ions enter into the amorphous area of cellulose and destroy the amorphous area which result in the cellulose I_C_ increment in agreement with Bodin et al. [53]. The I_C_ of the cellulose isolated from oil palm (*Elaeis guineensis*) empty fruit bunch (OPEFB)was 73.20% [28], while their values for pineapple leaves [54], indus-trial kelp (*Laminaria japonica*) [55], soy hulls [56], sisal, curaua, bamboo, and eucalyptus [57] and sugar palm (*Arenga Pinnata*) [58] were 54.00%, 69.40%, 73.50%, 78.00%, 87.00%, 87.00%, 89.00% and 85.90% respectively.

### 3.7. Effect of Test Temperature on I_C_

Figure 5 and Table 1 show that with increasing temperature, I_C_ increased for all nanocellulose types. It is shown that I_C_ for GNFC/H_2_SO_4_ was the highest followed by GNFC/NaOH, GNFC/NaClO, and RSNFC, in that order. Although Kusmono et al. [5] concluded that the crystallinity decreased as a result of the hydrolytic cleavage of the glycosidic bonds that caused due to the removed amorphous components and some crystalline parts at both the higher temperature and acid hydrolysis [59,60], Barbash et al. [18] found that upon temperature increment to 130 °C, I_C_ significantly increased from 30.94% to 59.30% due to the exerted homogenizer shear force exerted on the amorphous region of the cellulose fibers according to Zhao et al. [46]. Samir et al. [61] concluded that I_C_ increased to around 70.29% after alkaline hydrolysis and bleaching due to the significant decrement in the lignin content.

### 3.8. Effect of Reaction Time on I_C_

Figure 6 and Table 2 show that with increasing time, I_C_ increased for all nanocellulose types. The effect of the test periods on the values of I_C_ was lower for all specimen types compared to the effect of test temperatures in agreement with [5,18,49]. As Kargarzadeh et al. [59] and Kian et al. [62] concluded, a reaction time of more than 30 min resulted in crystallinity reduction, while Kargarzadeh et al. [59] found that for kenaf bast fibers NC production, the optimal reaction time was achieved at 40 min, 45 °C, and 65.00% sulfuric acid. But, the optimum hydrolysis time was achieved at 80 min with 58.00% sulfuric acid concentration according to Al-Dulaimi and Wanrosli [63]. 

### 3.9. Spectroscopic Analyses

Typical FT-IR spectra of RSNFC, GNFC/NaOH, GNFC/NaClO, and GNFC/H_2_SO_4_ are shown in Figure 7. Stretch - OH absorption peak at 3500 cm^−1^, associated - CH absorption peak at 2915 cm^−1^ and at 2850 cm^−1^ an overlapping of – CH had found in all samples. These peaks are only found in cellulosic feed stocks, while due to the amorphous cellulose chain termination RSNFC peaks have been lost in agreement with Zulnazri et al. [1].

Due to both the absorption of water and the strong interaction between the cellulose and air, a small absorption peaks at the range 1600–1650 cm^−1^ indicated that the cellulose samples no longer bind to O-H in agreement with Johar and Ahmad [64]. Associated with an aromatic ring polysaccharide, an absorption vibration band peak found at the range 1300–1350 cm^−1^ in agreement with that analyzed by Nacos et al. [65]. In addition, at region 1100 cm^−1^–1160 cm^−1^, absorption peaks were seen in agreement with Kagarzadeh et al. [59]. An intensity increment in the bands 1025 cm^−1^ due to the pyranose ring stretching occurred for all GNFCs treated types in agreement with Corrêa et al. [66] and absorption peaks 890 cm^−1^ related to C-H vibration of the lowest cellulose

All types of treated GNFCs showed increased intensity in the bands 1025 cm^−1^ due to the pyranose ring stretching in agreement with Corrêa et al. [66]. According to Li [67], absorption peaks of 890 cm^−1^ at C-H vibration band due to the lowest cellulose vibration anomeric, specifically to β-glucosides bonds that exist between the glucose units of cellulose/hemicellulose nanofibers in the GNFC spectra, [68]. According to the infrared spectral features, the functional types of cellulose nanofibers showed that there were no distinct changes at different conditions in agreement with [69].

### 3.10. Morphological Investigations of Untreated and Treated Fibers

The morphological changes of different types of different treated GNFC before and after sonication are shown in Figure 8. Due to the removal of hemicellulose and lignin and eliminating the cementing material around the fibers bundle, uniform fibers formed (Figure 8a). Thomas [26] concluded that acid hydrolysis removed the rest of the binding materials and highly ordered crystallites were formed. The fibrils formed aggregated and with rough surface morphology due to the effect H_2_SO_4_ acid hydrolysis in removing the cellulose amorphous components holding the cellulose crystal region that producing smoother GNFCs, (Figure 8b). Acid hydrolysis facilitates defibrillation of the fibers on a nanoscale level [60]. Figure 8c shows the morphology of the alkali-treated fibers on which the reduction of fibers diameters owing to the removing of some of the cementing material parts in agreement with [26].

Figure 8d shows the morphology of the alkaline sonicated GNFC. The sonication process causes further defibrillation due to the removal of the most lignin present in the GNFC fibers and the smaller of the bleached fibers compared to the untreated alkali fibers [61]. There are very distinct cellulose fiber bundles that exhibited a rough wood surface structure, Leite et al. [70] which proving that the acidic treatment is very effective for the lignin removal as well as the individual cellulose fibers separation.

Figure 8e shows GNFC/NaClO morphology before treatment by ultrasonic waves with lignin degradation which in turn facilitated the solubilization of lignin medium and separation according to Cherian et al. [54].

Figure 8f showed the morphology of GNFC/NaClO after sonication leading to further defibrillation, on which web-like nanostructure observed in most fibers exhibited web-like nanostructure with some bundles still existed [15,52].

### 3.11. Particle Size Measurement

Figure 9a–c showed the size distribution of GNFC/NaClO, GNFC/NaOH, and GNFC/H_2_SO_4_ measured by particle size analysis. The size distribution by volume of GNFC/NaClO as detected by laser diffraction shown in Figure 9a. There were two peaks; minor peak (1) with mean values of volume %, size (d·nm), and width of 3.20%, 173.20 d·nm, and 32.84 d·nm, respectively, and major peak (2) with a mean values of 96.80%, 13.23 d·nm, and 1.987 d·nm, respectively. Figure 9b shows GNFC/NaOH with only one peak with mean values of 100%, 104.1 d·nm, and 15.63 d·nm, while Figure 9c shows two peaks: the minor peak (1) with mean values of 87.9%, 582.8 d·nm, and 127.6 d·nm and with mean values of the major peak (2) of 12.1%, 102.1 d·nm, and 19.84 d·nm. The results showed that the produced weighted distribution volume contained different particle size measurements while that using image analysis resulted in the most accurate results according to [68,69,70,71]. Based on the given results, GNFC/NaClO produced better size distribution data compared to GNFC/NaOH and GNFC/H_2_SO_4_. Furthermore, the particle size distribution data proved that low acid concentration hydrolysis was not sufficient to obtain GNFC/H_2_SO_4_ in agreement with Mahardika et al. [72].

### 3.12. Zeta Potential Measurement

The Zeta potentials of the sonicated GNFC/NaClO, GNFC/NaOH, and GNFC/H_2_SO_4_ were measured and plotted in Figure 10. All specimen types showed a negative Zeta potential taking into consideration that the values lower than −15 mV represent the particle agglomeration starting and values higher than −30 mV indicated that sufficient mutual repulsion, resulting in a colloidal stability [73,74].

Figure 10 lists that the mean lowest negative value (−1.94 mv) was related to the sonicated GNFC/NaClO, while the mean highest value (−50.9 mv) was related to the sonicated GNFC/H_2_SO_4_ and sonicated GNFC/NaOH had a mean value of (−6.94 mv). Due to the presence of negatively charged sulfate groups on the cellulose nanocrystals surface, the sonicated GNFC/H_2_SO_4_ had a higher mean negative value according to Bondeson et al. [75]; Roman and Winter [76]. An aggregate form occurred due to the lack of electrostatic repulsive forces among the crystalline particles of the The sonicated GNFC prepared by NaClO hydrolysis in agreement with Araki et al. [77] and Angellier et al. [78]. Also, in aqueous media the use of H_2_SO_4_ reduces the starch nanocrystals agglomeration possibility and limits their flocculation.

Moreover, it was found that the conductivity values (mS/cm) for GNFC/NaClO, GNFC/NaOH, and GNFC/H_2_SO_4_ were 3.27 × 10^−4^, 2.12 × 10^−4^, and 1.68 × 10^−4^, respectively. As the lower conductivity leads to lower current values and lower stability [79,80], among the three suspensions, the Zeta potential was higher (GNFC/H_2_SO_4_ > GNFC/NaOH > GNFC/NaClO), indicating the higher acid concentration effect on the suspensions stability and the consequent colloidal suspension formation.

### 3.13. HPLC Nanocellulose Measurement

The nanocellulose samples prepared by the three methods were measured and compared with the Real Sample (RSNFC), shown in Figure 11a–d. They were found almost similar and at the same separation time.

## 4. Conclusions

Using three techniques: innovative method, acid, and alkaline hydrolysis carried out. GNFCs were produced from grass wastes via pretreatments using three treating mediums, NaClO, NaOH, and H_2_SO_4_, with optimum yield values of 95%, 90%, and 87%, respectively, at 25 °C, 20 min, and 5% concentration for each medium. The crystallinity index analyzed using the diffraction pattern and computed according to the Segal empirical method and the sum of the area under the crystalline adjusted peaks method. Both reaction temperature and time played an important role in the yielding and crystallinity index of GNFC. Reaction temperatures had a prominent effect on crystallinity index with optimum values of 40.705%, 70.489, 73.841, and 91.521 for sonicated RSNFC and sonicated GNFC treated with NaClO, NaOH, and H_2_SO_4_, respectively, at 80 °C, 20 min, and 5% concentration for each medium. Further augmentation of the GNFC surface charge occurred due to the ultrasonic homogenization. Both the morphological investigations of SEM and FT-IR resulted in untreated GNFC and treated GNFC with the three mediums found matched and in good consistence. The GNFCs were characterized for their size and surface by Zeta potential and HPLC. The isolated cellulose from the three treatment mediums compared with the standard sample (RSNFC) exhibited similar characteristics to those reported in the literature. 

## Figures and Tables

**Figure 1 polymers-14-01930-f001:**
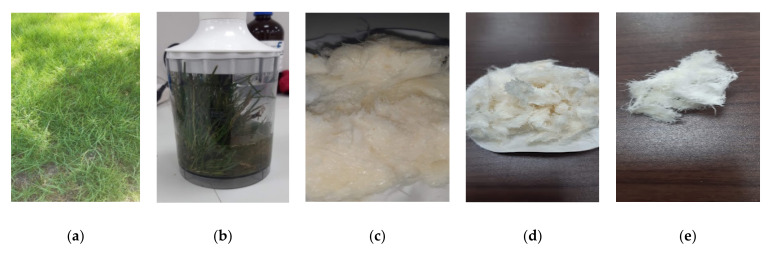
Preparation of cellulose from grass by sodium hypochlorite. (**a**) Type of grass, (**b**) mechanical cutting with NaClO, (**c**) after treatment, (**d**) after dying, and (**e**) after treatment by ultrasound waves.

**Figure 2 polymers-14-01930-f002:**
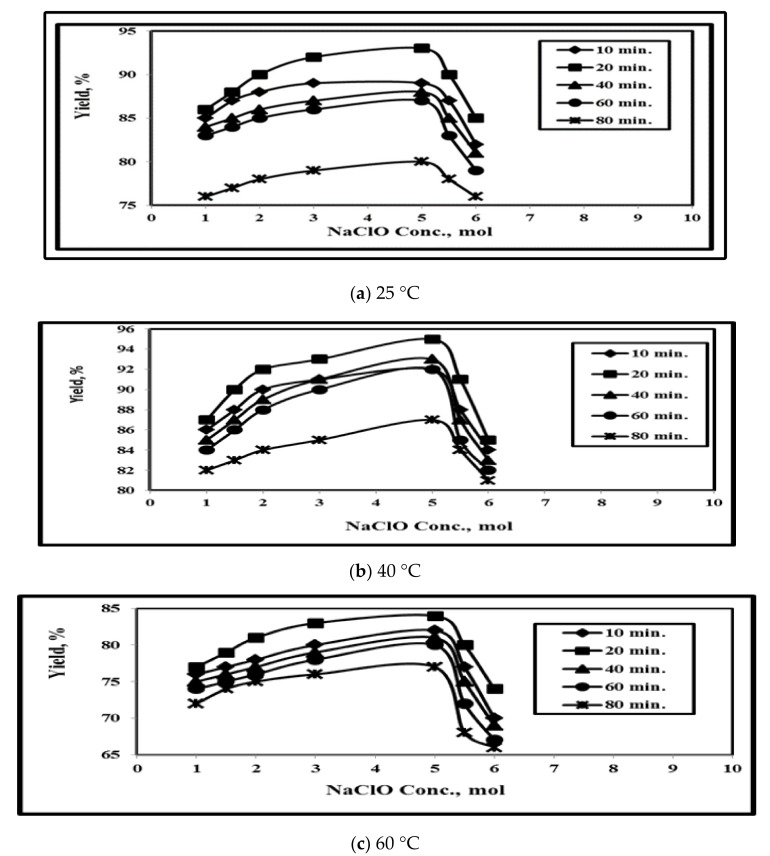
Effect of reaction time and NaClO concentration on the yield % at different test temperatures.

**Figure 3 polymers-14-01930-f003:**
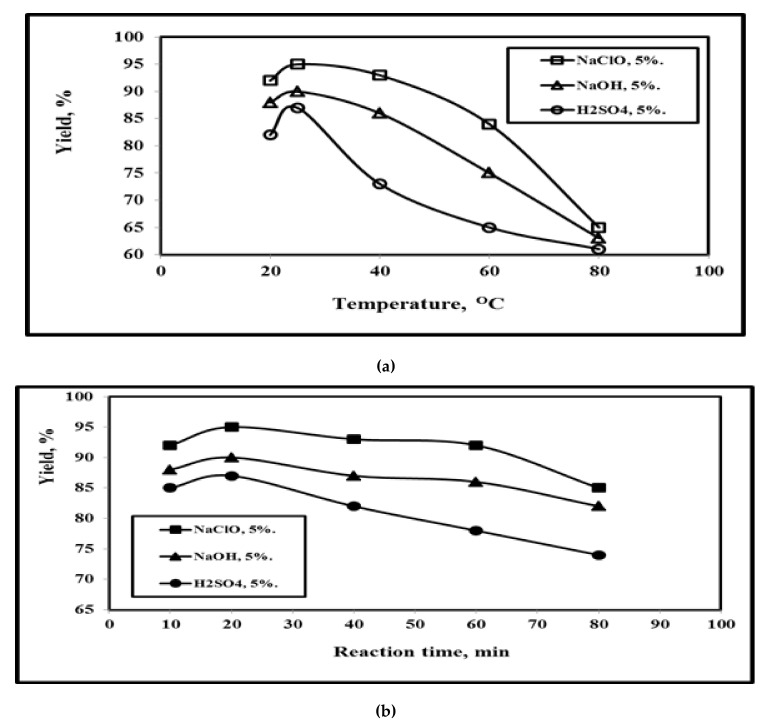
Effect of test temperatures and reaction times on the yield % of GNFC/5% NaClO, GNFC/5% NaOH, and GNFC/5% H_2_SO_4_.

**Figure 4 polymers-14-01930-f004:**
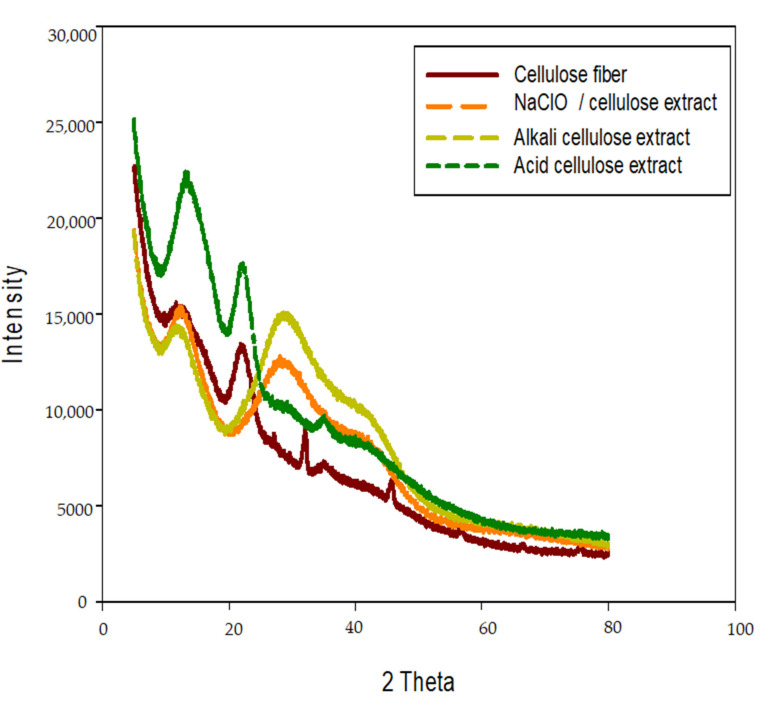
X-ray diffractions of cellulose fiber and nanocellulose after treatment by ultrasonic waves.

**Figure 5 polymers-14-01930-f005:**
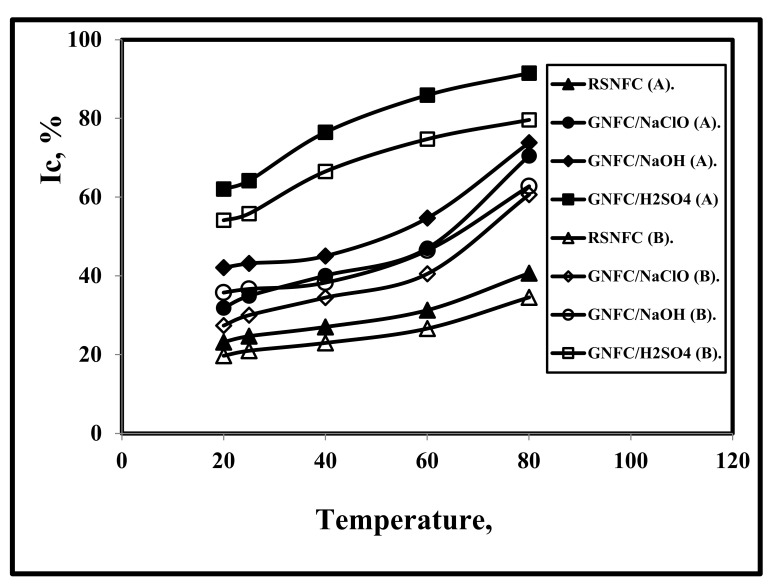
Effect of test temperature on the crystallinity of RSNFC, GNFC/5%NaClO, GNFC/5%NaOH, and GNFC/5%H_2_SO_4_ using both method A and method B.

**Figure 6 polymers-14-01930-f006:**
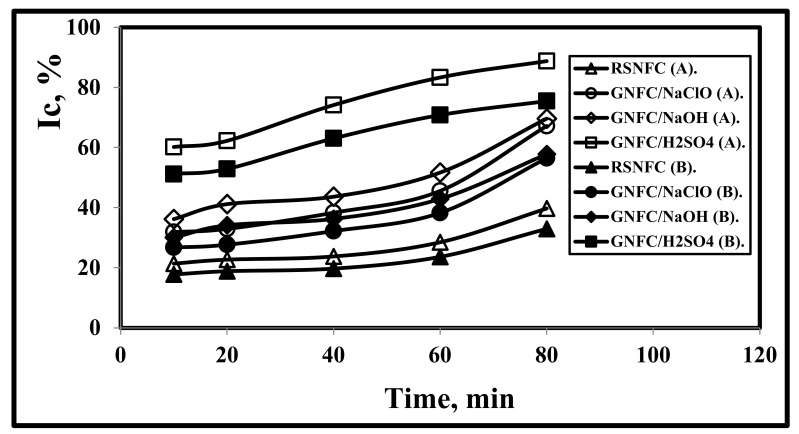
Effect of reaction time on the crystallinity of RSNFC, GNFC/5%NaClO, GNFC/5%NaOH, and GNFC/5%H_2_SO_4_ using both method A and method B.

**Figure 7 polymers-14-01930-f007:**
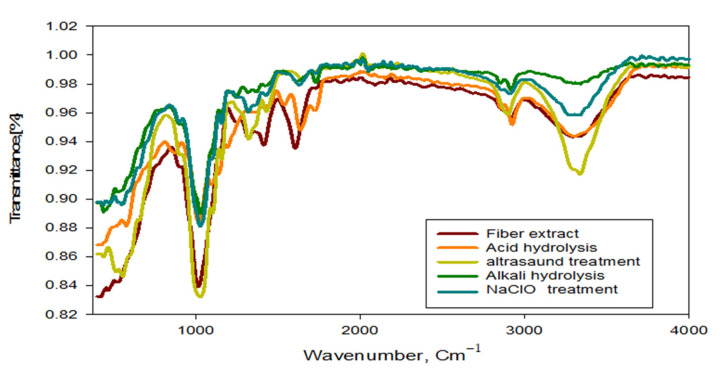
FTIR spectra of GNFC/5%NaClO, GNFC/5%NaOH and GNFC/5%H_2_SO_4_.

**Figure 8 polymers-14-01930-f008:**
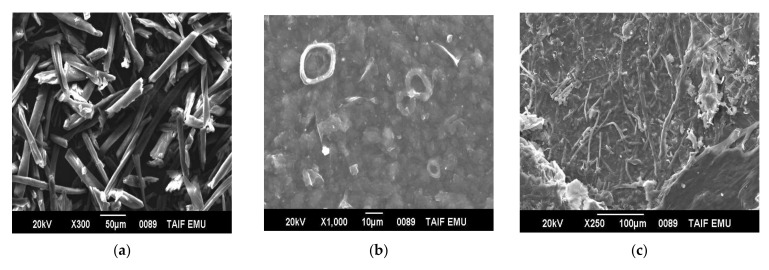
SEM of RSNFC; (**a**,**b**) acid hydrolysis before and after treatment by ultrasonic waves, (**c**,**d**) alkali hydrolysis before and after treatment by ultrasonic waves, and (**e**,**f**) for sodium hypochlorite before and after treatment by ultrasonic waves.

**Figure 9 polymers-14-01930-f009:**
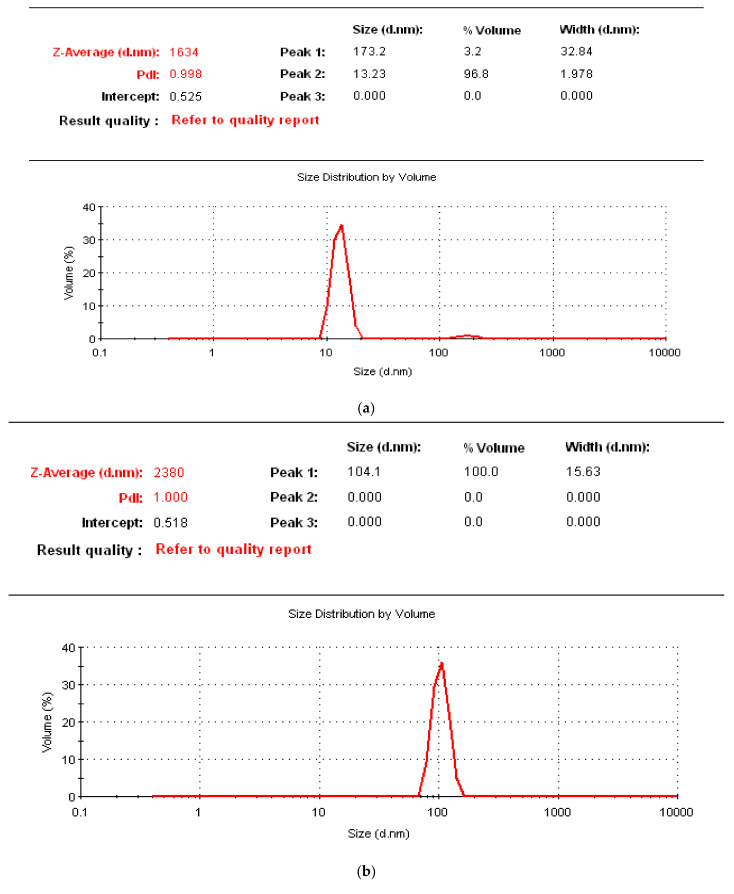
%Volume/size distribution of sonicated (**a**) GNFC/NaClO, (**b**) GNFC/NaOH, and (**c**) GNFC/H_2_SO_4_ using the particle size analysis (PSA).

**Figure 10 polymers-14-01930-f010:**
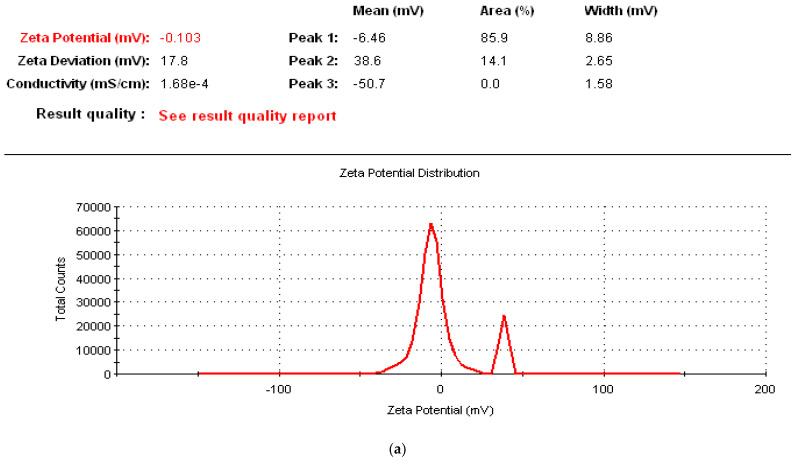
Surface Zeta potential of nanocellulose GNFC/H_2_SO_4_, GNFC/NaClO, and GNFC/NaOH hydrolysis. (**a**) Surface Zeta potential of nanocellulose GNFC/H_2_SO_4_ hydrolysis. (**b**) Surface Zeta potential of GNFC/NaClO hydrolysis. (**c**) Surface Zeta potential of GNFC/NaOH hydrolysis.

**Figure 11 polymers-14-01930-f011:**
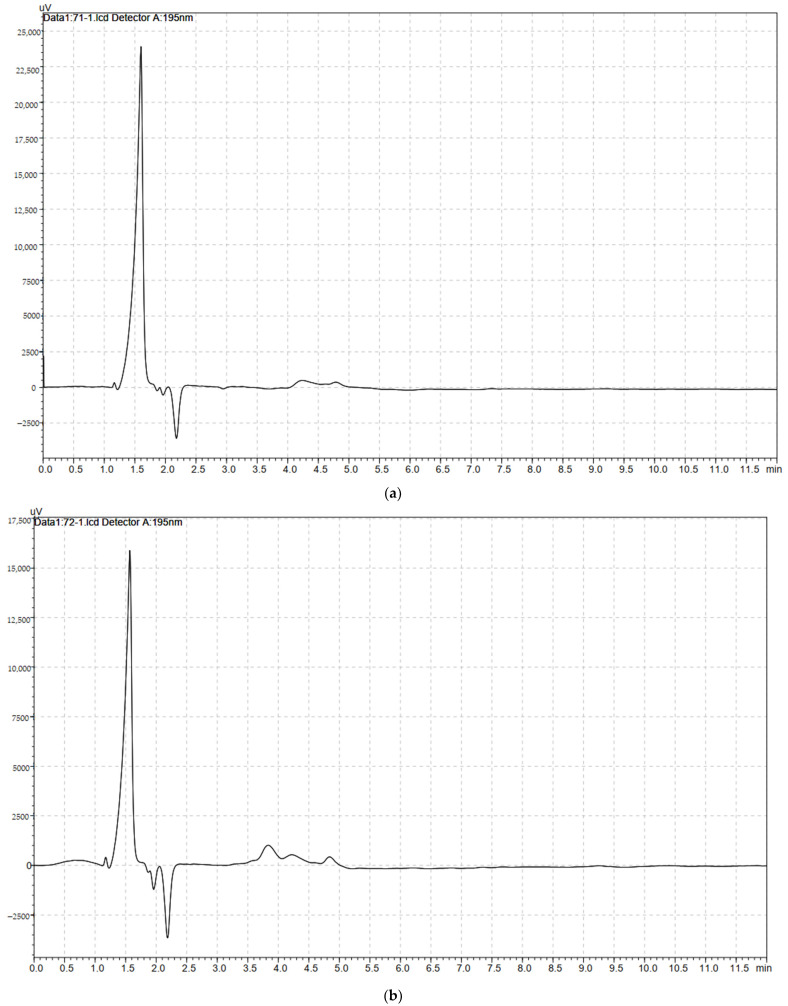
HPLC (**a**) standard nanocellulose (RSNFC), (**b**) nanocellulose GNFC/H_2_SO_4_, (**c**) GNFC/NaClO, and (**d**) GNFC/NaOH hydrolysis.

**Table 1 polymers-14-01930-t001:** Variation of I_C_ for RSCNFC (Real Sample Commercial Nanofiber Cellulose), GNFC/NaClO, GNFC/NaOH, and GNFC/H_2_SO_4_ with test temperature using both method A and method B.

Temp. °C	Method (A)				Method (B)			
	I_C_% RSNFC	I_C_% (GNFC/NaClO)	I_C_% (GNFC/NaOH)	I_C_% (GNFC/H_2_SO_4_)	I_C_% RSNFC	I_C_% (GNFC/NaClO)	I_C_% (GNFC/NaOH)	I_C_% (GNFC/H_2_SO_4_)
20	23.205	31.853	42.095	62.083	19.724	27.394	35.781	54.122
25	24.69	34.93	43.16	64.17	20.987	30.039	36.686	55.828
40	27.056	40.095	45.074	76.477	22.998	34.482	38.313	66.535
60	31.303	47.04	54.689	85.899	26.608	40.544	46.486	74.732
80	40.705	70.489	73.841	91.521	34.559	60.621	62.765	79.623

**Table 2 polymers-14-01930-t002:** Variation of I_C_ for RSCNFC (Real Sample Commercial Nanofiber Cellulose), GNFC/NaClO, GNFC/NaOH, and GNFC/H_2_SO_4_ with reaction time using both method A and method B.

Timemin	Method (A)				Method (B)			
	I_C_% RSNFC	I_C_% (GNFC/NaClO)	I_C_% (GNFC/NaOH)	I_C_% (GNFC/H_2_SO_4_)	I_C_% RSNFC	I_C_% (GNFC/NaClO)	I_C_% (GNFC/NaOH)	I_C_% (GNFC/H_2_SO_4_)
10	21.401	31.853	36.22	60.221	17.763	26.757	30.063	51.188
20	22.69	32.93	41.16	62.245	18.833	27.661	34.163	52.908
40	23.745	38.33	43.675	74.183	19.708	32.197	36.250	63.056
60	28.494	45.584	51.678	83.322	23.650	38.291	42.893	70.824
80	39.701	67.194	69.661	88.775	32.952	56.443	57.819	75.459

## Data Availability

The author confirms that the data of this study are available within the article.

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
