# Peer review of "An Innovative Preparation, Characterization, and Optimization of Nanocellulose Fibers (NCF) Using Ultrasonic Waves"

_polymers, 2022, doi:10.3390/polym14101930_

Round 1

Reviewer 1 Report

  1. The abstract is to be more concise.
  2. In the Introduction the author lists lots of previous works, but there is no clear evidence how do they help to understand the achievement of the present work and what kind of advance against that works is done.
  3. There is no reference to Fig. 1 in the text.
  4. There are no reference XRD and FTIR spectra which would add clarity to data interpretation, especially when plotted against each other for different samples.
  5. The logical connection and complementarity of results of different techniques in terms of achieving the goal of the study is to be gained.
  6. The Conclusion lists results coming out from specific measurements. In my view, it needs a statement that highlights the main message of the work.
  7. There are numerous misleading statements throughout the manuscript. To name a few:

“In all previous researches, nanocellulose prepared by acid decomposition or alkaline decomposition, after going through several long and expensive steps.”, I guess is grammatically incorrect or incomplete.

 “In my research, an innovative, environmentally friendly and economically way to prepare nanocellulose from grass as vegetable waste from large areas of playgrounds, parks and public gardens with vegetation cover that includes grass.”

“Zhenghao et al. [4] concluded that cellulose, lignin and lignocellulose are important bioresources in the nature due to their effective and environmentally friendly utilization not only protects the environment but also reduces dependence on fossil resources.”

“Chen et al. [8] showed that XRD profiles of Cr(NO3)3 hydrolysis to isolate cellulose nanocrystals (CNCCr(NO3)3) had the major peaks at around 2θ = 15.1o (1Ä«0), 16.5o (110), 22.5o (200), and 34.6° (004), which indicated the presence of cellulose I structure.”

“Cellulose fibers were isolated from gardens grass was mowed in large quantities periodically from the vegetation cover of large areas of stadium floors, parks and public gardens.”

“In my method, the collected grass washed with rinsed water and grinded mechanically with sodium hypochlorite of different concentrations; 1, 1.5, 2.5 and 5.0 mol for 10, 20, 40, 60, and 80 min at 25, 40, 60, and 80oC.”

“The yield of GNFC affected by reaction temperature as shown in Fig. 2.”

“The partial removal of impurities associated with the GNFC/5% H2SO4 led to a slightly higher intensity of the XRD peaks while chemical pretreatments by alkali hydrolysis and bleaching were noted to be an effective method to purify cellulose in the autoclaved sample and the sharp peaks with the markedly higher intensity, of the chemically treated samples indicated a higher purity of cellulose due to the more efficient removal of noncellulosic materials and dissolution of the amorphous region in agreement with Khukutapan [15].”,  - an extremely long sentence.

Author Response

REVIEWER

No.

REVIEWER COMMENT

Corrections of Reviewer Comments

Reviewer No. (1)

English language and style:

(x) Extensive editing of English language and style required

a.Does the introduction provide sufficient background and include all relevant references.

Must be improved.

b.Is the research design appropriate?

Must be improved.

c.Are the methods adequately described?

Must be improved.

d.Are the results clearly presented?

Must be improved.

e.Are the conclusions supported by the results?

Can be improved.

TheEnglish language and style are corrected according to Reviewer (1) commentin red.

a.The introduction improved according to the reviewer commentin red.

b.The research design improved according to Reviewer (1)in red.

c.The description of the method improved according to Reviewer (1) commentin red..

d.The presented results improved according to Reviewer (1) commentin red.

e.The conclusions improved according to Reviewer (1) commentin red.

1.The abstract is to be more concise.

1.The ABSTRACT reduced from 368 words to 247 words to make the ABSTRACT concise in red.

2.In the Introduction the author lists lots of previous works, butthere is no clear evidence how do they help to understand theachievement of the present work and what kind of advanceagainst that works is done.

2.In the Introduction, the First seven references illustrated the cellulose characteristics, usages, sources and its ways of dependence. References 8-14 illustrated the previous results of the cellulose X-Ray results which is important to my work. References 15-18 illustrated the crystallinity index and yield values of the previous investigators cellulose sources which will be comparable to my work. References 19-30 illustrated recent advances in the preparation, modification, and emerging application of nanocellulose of the latest investigations of the past 3 years in which my work must be compared.

3.There is no reference to Fig. 1 in the text.

3.Fig. 1 was added under the title: “2.4. Innovative Preparation of Grass Nano Fiber Cellulose (IGNFC)”.

4.There are no reference XRD and FTIR spectra which wouldadd clarity to data interpretation, especially when plottedagainst each other for different samples.

4.The XRD and FTIR spectra figures are added at page 13 and page 18 respectively.

5. The logical connection and complementarity of results ofdifferent techniques in terms of achieving the goal of the studyis to be gained.

5.The logical connection and complementarity of results ofdifferent techniques in terms of achieving the goal of the studyis carried out and discussed.

6.The Conclusion lists results coming out from specificmeasurements. In my view, it needs a statement that highlightsthe main message of the work.

6.The Conclusion lists results coming out from specificmeasurements and the statements highlightsthe main message of the work.

7.There are numerous misleading statements throughout the manuscript. To name a few:

7.a.“In all previous researches, nanocellulose prepared by acid decomposition or alkaline decomposition, after going through several long and expensive steps.”, I guess is grammatically incorrect or incomplete.

7.b. “In my research, an innovative, environmentally friendly and economically way to prepare nanocellulose from grass as vegetable waste from large areas of playgrounds, parks and public gardens with vegetation cover that includes grass.”

7.c.“Zhenghao et al. [4] concluded that cellulose, lignin and lignocellulose are important bioresources in the nature due to their effective and environmentally friendly utilization not only protects the environment but also reduces dependence on fossil resources.”

7.d. “Chen et al. [8] showed that XRD profiles of Cr(NO3)3 hydrolysis to isolate cellulose nanocrystals (CNCCr(NO3)3) had the major peaks at around 2θ = 15.1o (1Ä«0), 16.5o (110), 22.5o (200), and 34.6°(004), which indicated the presence of cellulose I structure”.

7.e.“Cellulose fibers were isolated from gardens grass was mowed in large quantities periodically from the vegetation cover of large areas of stadium floors, parks and public gardens.”

7.f. “In my method, the collected grass washed with rinsed water and grinded mechanically with sodium hypochlorite of different concentrations; 1, 1.5, 2.5 and 5.0 mol for 10, 20, 40, 60, and 80min at 25, 40, 60, and 80oC.”

7.g. “The yield of GNFC affected by reaction temperature as shown in Fig. 2”.

7.h.“The partial removal of impurities associated with the GNFC/5%H2SO4 led to a slightly higher intensity of the XRD peaks while chemical pretreatments by alkali hydrolysis and bleaching were noted to be an effective method to purify cellulose in the autoclaved sample and the sharp peaks with the markedly higher intensity, of the chemically treated samples indicated a higher purity of cellulose due to the more efficient removal of noncellulosic materials and dissolution of the amorphous region in agreement with Khukutapan [15].”,- an extremely long sentence.

7.The misleading statements throughout themanuscript are corrected.

7.a. This sentence is deleted.

7.b.This research is innovative, environmentally friendly and economically way to prepare nanocellulose since cellulose is obtained from natural material (grass) vegetable waste that found in large areas of playgrounds, parks and public gardens with vegetation cover that includes grass.”

7.c.This statement is true, since cellulose can be extracted from the crude oil polymers. So, finding other cellulose resources reduced the dependence on crude oil resources (fossil resources).

7.d.According to data of all experts and guides in XRD of cellulose, the natural cellulose has XRD peaks in 2Ï´ ranges of 14.5-15o, 16-16.5o, 22.4-22.7o and 34-35.5o if wavelength of X-ray is 0.1542 nm, but these values are related to Chen et al. [8].

7.e.In my work, the cellulose fibers are obtained from the vegetation cover of large areas of stadium floors, parks and public gardens.

7.f.In my work, the grass cellulose is prepared through these steps and the test variables (concentration, time and temperature) are chosen with respect to the previous investigators to make a real comparison with their results.

7.g. Yes, Fig. 2. a-d found in pages 9-10 showed the effect of reaction temperatures on the GNFC.

7.h. This sentence is reduced in page 12 according to Reviewer (1) comment.

Reviewer No. (2)

English language and style:

(x) Extensive editing of English language and style required

a. Does the introduction provide sufficient background and include all relevant references.

YES.

b.Is the research design appropriate?

YES.

c.Are the methods adequately described?

Can be improved.

d.Are the results clearly presented?

YES.

e.  Are the conclusions supported by the results?

Can be improved.

TheEnglish language and style are corrected according to Reviewer (1) commentin red.

a.Thanks to Reviewer (2).

b.Thanks to Reviewer (2).

c.The description of the method improved according to Reviewer (1) commentin red..

d.Thanks to Reviewer (2).

e.The conclusions improved according to Reviewer (1) commentin red.

1. The introduction is very interesting and makes a summary of articles described in the literature on nanocellulose.

1. Thanks for Reviewer (2)

2.Revise and rewrite the Part: Effect of Reaction Medium on GNFC Yield”.

2. The Effect of Reaction Medium on GNFC Yield on GNFC Yield is revised and corrected according to Reviewer (2) comment in red.

3. Revise the presentation the Part “Zeta potential measurements”.

3.The Part “Zeta potential measurements” is revised and corrected according to Reviewer (2) comment in red.

4. Revise the presentation of “Conclusions”, are summary of paper.

4. Both the Conclusions and paper summary presentation are revised and corrected according to Reviewer (2) comment in red.

5. Page 2, Line 1, change GNFC/ NaOH, GNFC/ NaClO for GNFC/NaOH, GNFC/NaClO.

5. It is corrected according to Reviewer (2) comment in red.

6. Page 2, Line 2, define the letters “RSNFC”. Revise all manuscript.

6. It is defined corrected according to Reviewer (2) comment in red.

7. Page 2, Line 2, delete “respectively”.

7. It is deleted corrected according to Reviewer (2) comment in red.

8. INTRODUCTION

8.a. Page 3, Line 22, change Cr(NO3)3 for Cr(NO) and CNCCr(NO)for CNCCr(NO3) 3.

8.b.Page 3, Line 25-26, change 16.389º for 16.39º and 15.0º for15.00º, Revise all manuscript.

8.a.It is changed corrected according to Reviewer (2) comment in red.

8.b. It is changed corrected according to Reviewer (2) comment in red.

9. EXPERIMENTAL

9.a.Page 5, Line 10, change Acetic anhydride for anhydride acetic acid.

9.b.Page 6, Line 5, delete “which”.

9.c.Page 6, Line 9, change “for20 min” for “for 20 min”.

9.d.Page 6, Line 10, change 14000 for 14,000.

9.e.Page 6, Line 10, change ultra- sonication for ultra-sonication.

9.f.Page 6, Lie 11, change further use, for further use. Revise all manuscript.

9.g.Figure 1, change Na hypochlorite for NaClO. Revise the formulas in all manuscript.

9.a. It is changed corrected according to Reviewer (2) comment in red.

9.b. It is deleted corrected according to Reviewer (2) comment in red.

9.c. It is changed corrected according to Reviewer (2) comment in red.

9.d.It is changed corrected according to Reviewer (2) comment in red.

9.e.It is changed corrected according to Reviewer (2) comment in red.

9.f. It is changed corrected according to Reviewer (2) comment in red.

9.g.It is changed corrected according to Reviewer (2) comment in red.

10. CHARACTERIZATIONS

10.a. Page 7, Line 3, change Perki-nElmer for Perkin-Elmer.

10.b.Page 7, Line 7, Line 9, change 60 ºC for 60ºC. Write the same form in all manuscript.

10.c.Page 8, Line 7, change 750mg for 750 mg

10.d. 10.d. Page 8, Line 8, change Cu Ka for Cu Kα.

10.e. Page 10, Line 16, change of120 rpm for of 120 rpm.

10.f. Line 19, change NPswas for NPs was.

10.g. Page 10, Line 19, change 5%HSO for 5% H2SO4.

10.a. It is changed corrected according to Reviewer (2) comment in red.

10.b. It is changed corrected according to Reviewer (2) comment in red.

10.c.It is changed corrected according to Reviewer (2) comment in red.

10.d. It is changed corrected according to Reviewer (2) comment in red.

10.e. It is changed corrected according to Reviewer (2) comment in red.

10.f. It is changed corrected according to Reviewer (2) comment in red.

10.g.It is changed corrected according to Reviewer (2) comment in red.

11. RESULTS and DISCUSSIONS

11.a. Page 11, Line 6, 5% M? Which is the compound?.

11.b. Page 12, Line 17, change HSO respectively for H2SOrespectively. Revise all manuscript.

11.c. Page 12, Line 17-18, delete “with temperature increment”.

11.d. Page 15, Line 4, change Table 1. And Table 2 for Table 1 and table2

11.e.Revise the presentation of the Table 1.

11.f. Page 17, Line 5, change 45 for 45º.

11.g. Page 19, Line 4, change -CH2 for -CH. Revise all manuscript.

11.h. Page 19, Line 4, change cm-1 for cm. Revise all manuscript.

11.i.Page 19, Line 8, change 1600-1650 cm for 1650-1600 cm-1.

11.j. Revise all manuscript.

11.k. Page 22, Line 23, change which that reported for which reported.

11.a. The compound is added according to Reviewer (2) comment in red.

11.b.It is changed  and revised according to Reviewer (2) comment in red.

11.c. It is deletedaccording to Reviewer (2) comment.

11.d. It is changed  and revised according to Reviewer (2) comment in red.

11.e. It is revised according to Reviewer (2) comment in red.

11.f. It is changed  and revised according to Reviewer (2) comment in red.

11.g. It is changed  and revised according to Reviewer (2) comment in red.

11.h. It is changed  and revised according to Reviewer (2) comment in red.

11.i.It is changed according to Reviewer (2) comment in red.

11.j. The manuscript revised according to Reviewer (2) comment in red.

11.k.It is changed according to Reviewer (2) comment in red.

Reviewer 2 Report

Polymers-1506657

Environmental and ecological concerns have a great interest in this moment, high cost and expansion of petroleum-based synthetic materials. A product has gained further attention is the Nanocellulose. In this paper, an innovative environmentally friendly and economically process for the obtention of the nanocellulose is described.

In my opinion this manuscript is of sufficiently quality for publication in Polymers, but is necessary that the author review of manuscript

Comment

  • The introduction is very interesting and makes a summary of articles described in the literature on nanocellulose
  • Revise and rewrite the Part: Effect of Reaction Medium on GNFC Yield”
  • Revise the presentation the Part “Zeta potential measurements”
  • Revise the presentation of “Conclusions”, are summary of paper

Other comment

  • Page 2, Line 1, change GNFC/ NaOH, GNFC/ NaClO for GNFC/NaOH, GNFC/NaClO
  • Page 2, Line 2, define the letters “RSNFC”. Revise all manuscript
  • Page 2, Line 2, delete “respectively”

Introduction,

  • Page 3, Line 22, change Cr(NO3)3 for Cr(NO3)3 and CNCCr(NO3)3 for CNCCr(NO3)3
  • Page 3, Line 25-26, change 16.389º for 16.39º and 15.0º for 15.00º, Revise all manuscript

Experimental

  • Page 5, Line 10, change Acetic anhydride for anhydride acetic acid
  • Page 6, Line 5, delete “which”
  • Page 6, Line 9, change “for20 min” for “for 20 min”
  • Page 6, Line 10, change 14000 for 14,000
  • Page 6, Line 10, change ultra- sonication for ultra-sonication
  • Page 6, Lie 11, change further use, for further use. Revise all manuscript
  • Figure 1, change Na hypochlorite for NaClO. Revise the formulas in all manuscript

Characterizations

  • Page 7, Line 3, change Perki-nElmer for Perkin-Elmer
  • Page 7, Line 7, Line 9, change 60 ºC for 60ºC. Write the same form in all manuscript
  • Page 8, Line 7, change 750mg for 750 mg
  • Page 8, Line 8, change Cu Ka for Cu Kα
  • Page 10, Line 16, change of120 rpm for of 120 rpm
  • Page 10, Line 19, change NPswas for NPs was
  • Page 10, Line 19, change 5%H2SO4 for 5% H2SO4

Results and Discussion

  • Page 11, Line 6, 5% M? Which is the compound?
  • Page 12, Line 17, change H2SO4 respectively for H2SO4, respectively. Revise all manuscript
  • Page 12, Line 17-18, delete “with temperature increment”
  • Page 15, Line 4, change Table 1. And Table 2 for Table 1 and table 2
  • Revise the presentation of the Table 1
  • Page 17, Line 5, change 45 for 45º
  • Page 19, Line 4, change -CH2 for -CH2. Revise all manuscript
  • Page 19, Line 4, change cm-1 for cm-1. Revise all manuscript
  • Page 19, Line 8, change 1600-1650 cm-1 for 1650-1600 cm-1­. Revise all manuscript
  • PAGE 22, Line 23, change which that reported for which reported

Author Response

(The authors gave the same response as above.)

Round 2

Reviewer 1 Report

Manuscript is improved.

Author Response

thanks very very match
